# SAGMAN: STABILITY ANALYSIS OF GRAPH NEURAL NETWORKS (GNNS) ON THE MANIFOLDS

## ABSTRACT

Graph neural networks (GNNs) are highly effective at tasks that involve analyzing graph-structured data, such as predicting protein interactions, modeling social networks, and identifying communities within graphs. However, modern GNNs can be sensitive to changes in the input graph structure and node features, leading to unpredictable behavior and degraded performance. In this work, we introduce a spectral framework (SAGMAN) for analyzing the stability of GNNs. SAGMAN quantifies the stability of each node by examining the distance mapping distortions (DMDs) on the input/output manifolds: when two nearby nodes on the input manifold are mapped (through a GNN model) to two distant nodes (data samples) on the output manifold, it implies a large DMD and thus poor GNN stability. To create low-dimensional input/output manifolds for meaningful DMD estimations while exploiting both the input graph topology and node features, we propose a spectral sparsification framework for estimating probabilistic graphical models (PGMs) such that the constructed input/output graph structures can well preserve pair-wise distances on the manifolds. Our empirical evaluations show that SAGMAN can effectively reveal each node's stability under various edge/feature perturbations, offering a scalable approach for assessing the stability of GNNs.

## 1 INTRODUCTION

The advent of Graph Neural Networks (GNNs) has sparked a significant shift in machine learning (ML), particularly in the realm of graph-structured data (Keisler, 2022; Hu et al., 2020; Kipf & Welling, 2016; Veličković et al., 2017; Zhou et al., 2020). By seamlessly integrating graph structure and node features, GNNs yield low-dimensional embedding vectors that maximally preserve the graph structural information (Grover & Leskovec, 2016). Such networks have been successfully deployed in a broad spectrum of real-world applications, including but not limited to recommendation systems (Fan et al., 2019), traffic flow prediction (Yu et al., 2017), chip placement (Mirhoseini et al., 2021), and social network analysis (Ying et al., 2018). However, the enduring challenge in the deployment of GNNs pertains to their stability, especially when subjected to perturbations in the graph structure (Sun et al., 2020; Jin et al., 2020; Xu et al., 2019). Recent studies suggest that even minor alterations to the graph structure (encompassing the addition, removal, or rearrangement of edges) can have a pronounced impact on the performance of GNNs (Zügner et al., 2018; Xu et al., 2019). This phenomenon is particularly prominent in tasks such as node classification (Yao et al., 2019; Veličković et al., 2017; Bojchevski & Günnemann, 2019). The concept of stability here transcends mere resistance to adversarial attacks, encompassing the network's ability to maintain consistent performance despite inevitable variations in the input data (graph structure and node features).

In the literature, there are a few studies attempting to analyze GNN stability. Specifically, (Keriven et al., 2020) first studied the stability of graph convolutional networks (GCN) on random graphs under small deformation. Later, (Gama et al., 2020) and (Kenlay et al., 2021) explored the robustness of various graph filters, which are then used to measure the stabilities of the corresponding (spectral-based) GNNs. However, these prior methods are limited to either synthetic graphs or specific GNN models.

In this work, we present SAGMAN, a novel framework devised to quantify the stability of GNNs through individual nodes. This is accomplished by assessing the Distance Mapping Distortions

(DMDs) of node pairs within a given (pre-trained) GNN model. Specifically, we present an efficient method to represent the input graph (including node features) and GNN logits as graph-based manifolds. Consequently, DMD assesses the distortion of pairwise distances between nodes by GNN from the input to the output manifold. For constructing graph-based manifolds, SAGMAN integrates the PGM with spectral embedding. This approach learns the manifold topology while retaining essential structural (spectral) properties of the input graph, such as pairwise resistance distances. Importantly, SAGMAN ensures nearly-linear algorithmic complexity, functioning on both homophilic and heterophilic graphs. Its data-centric nature allows operation across various GNN variants, independent of label information, network architecture, and learned parameters, highlighting SAGMAN's extensive applicability. Experimental results affirm SAGMAN's capability in gauging individual node stability within diverse GNN models on realistic graphs, guiding more effective adversarial target attacks. It is crucial to note that this study aims not to bolster GNN stability but to offer significant insights for refining the stability and robustness of GNNs. The key technical contributions of this work are outlined below:

• To the best of our knowledge, we are the first to propose a spectral method for evaluating node-level stability in GNNs, which is achieved via measuring the DMD of adjacent nodes on input/output graph-based manifolds.

• To construct proper graph-based manifolds for estimating DMDs, SAGMAN exploits PGMs and spectral graph embedding to learn low-dimensional manifolds while preserving key structural properties of the given graph dataset. Our empirical results validate that our approach provides a scalable method for assessing individual node stability across various GNN models in realistic graphs and guiding more potent adversarial target attacks.

• SAGMAN has a near-linear time complexity and is agnostic to GNN models as well as node label information. Thus, SAGMAN is very flexible to various sizes of graphs, types of GNNs, and node-level tasks.

## 2 BACKGROUND

### 2.1 SPECTRAL GRAPH THEORY

Spectral graph theory is an area of study in mathematics that explores the relationship between a graph's structural properties and its matrices' characteristics, particularly the graph Laplacian (Chung, 1997). The graph Laplacian matrix, denoted as $L$, is defined as $L = D - A$, where $D$ is the degree matrix and $A$ is the adjacency matrix. The eigenvalues and eigenvectors of the Laplacian matrix, also known as the spectrum of the graph, encode essential topological and geometric properties of the graph. The spectrum can shed light on various structural aspects of the graph, such as the number of connected components, the existence of a spanning tree, and the graph's diameter, among others. Seminal works (Bruna et al., 2013; Defferrard et al., 2016) highlight the potential of spectral methods for graph-based learning tasks.

### 2.2 PROBABILISTIC GRAPHICAL MODEL (PGM)

PGMs, also known as Markov Random Fields (MRFs), are powerful tools in machine learning and statistical physics for representing complex systems with intricate dependency structures (Roy et al., 2009). In recent years, methods based on graph signal processing (GSP) have been extensively explored for their efficiency in estimating PGMs from data samples (Dong et al., 2019; Lake & Tenenbaum, 2010). If we consider each column vector in the data matrix $X \in \mathbb{R}^{N \times M}$ as a graph signal vector, the maximum likelihood estimation (MLE) of the precision matrix $\Theta$ (PGM) can be obtained by solving the following convex optimization problem (Dong et al., 2019):

$$\max_{\Theta} : F(\Theta) = \log \det(\Theta) - \frac{1}{M} Tr(X^{\top} \Theta X) \tag{1}$$

where $\Theta = L + \frac{1}{\sigma^2} I$, $Tr(\bullet)$ denotes the matrix trace, $L$ denotes the set of undirected graph Laplacian matrices, $I$ denotes the identity matrix, and $\sigma^2 > 0$ denotes prior feature variance.

## 2.3 STABILITY OF GNNs

For GNNs, stability is a fundamental aspect that warrants in-depth analysis and comprehension. The stability of a GNN refers to its performance (output) stability in the presence of edge/node perturbations (Sun et al., 2020). This includes the ability to maintain the fidelity of predictions and outcomes when subjected to changes such as edge alterations or feature attacks.

To elucidate, consider the scenario where the input graph data is subject to minor modifications, such as the addition or removal of a node, a change in node features, or a slight alteration in the graph structure. A desired GNN model is expected to exhibit good stability, wherein every predicted output or the graph embeddings do not change drastically in response to the aforementioned minor perturbations (Jin et al., 2020; Zhu et al., 2019). On the contrary, an unstable GNN would exhibit a considerable change in part of outputs or embeddings, potentially leading to erroneous inferences or predictions.

The importance of stability in GNNs has been underlined by several recent studies. For instance, the vulnerability of GNNs to adversarial attacks has been demonstrated, where small yet carefully crafted perturbations on targeted nodes can lead to significant misclassifications (Zügner et al., 2018). This underpins the importance of ensuring stability in GNNs against such adversarial scenarios. Consequently, quantifying sample stability in GNNs is key to assessing their performance and stability. Although there are a few prior studies exploring GNN stability, they are restricted to either synthetic graphs or specific GNN models (Keriven et al., 2020; Gama et al., 2020; Kenlay et al., 2021). More importantly, these approaches are not aimed at developing a metric to quantify GNN stability. In contrast, this work proposes a stability metric that allows for characterizing the consistency of the model's prediction or representation for a specific sample (e.g., a node) when exposed to perturbations.

## 2.4 DISTANCE MAPPING DISTORTION (DMD)

The stability of an ML model fundamentally refers to the ability of the model to produce consistent output despite small variations or noise in the input (Szegedy et al., 2013). Given this definition, a natural metric for quantifying the stability of ML models has been derived from the DMD concept (Cheng et al., 2021). DMD provides an effective measure to evaluate how an ML model (function) distorts the pairwise distances between data samples. Specifically, let $M$ denote the ML model, which operates on input $X$ to yield output $Y$, i.e., $Y = M(X)$. The DMD between samples $p$ and $q$, denoted as $\delta^M(p, q)$, is defined as the ratio of the distance $d_Y(p, q)$ in the output manifold $Y$ to the distance $d_X(p, q)$ in the input manifold $X$ (Cheng et al., 2021). Formally, the DMD is computed as follows (Cheng et al., 2021):

$$\delta^M(p, q) \stackrel{\text{def}}{=} \frac{d_Y(p, q)}{d_X(p, q)}. \tag{2}$$

The geodesic distance metric is arguably the most natural choice to calculate DMD on the manifolds (Cheng et al., 2021; Naitzat et al., 2020). However, computing all pairs geodesic distance between $N$ input (output) data points can be prohibitively expensive even when taking advantage of the state-of-the-art randomized algorithms (Williams, 2018). To avoid the staggering computational cost, effective-resistance distances have been adopted for graph-based manifolds (Chen et al., 2021). The effective-resistance distance $d^{eff}(p, q)$ between any two nodes $p$ and $q$ for an undirected and connected graph $G = (V, E)$ satisfies (Cheng et al., 2021):

$$d^{eff}(p, q) = e_{p,q}^\top L_G^+ e_{p,q} = \|U_N^\top e_{p,q}\|_2^2 \tag{3}$$

where $e_{p,q} = e_p - e_q$, $e_p \in \mathbb{R}^N$ denotes the standard basis vector with the $p$-th element being 1 and others being 0, $L_G^+ \in \mathbb{R}^{N \times N}$ denotes the Moore–Penrose pseudoinverse of the graph Laplacian matrix $L_G \in \mathbb{R}^{N \times N}$, $U_N$ denotes the eigensubspace matrix including $N - 1$ nontrivial weighted Laplacian eigenvectors, more details regarding $U_N$ is in Appendix A.2.

The maximum LHS of Equation 1 can be represented using resistance distance as $\max_{\forall p,q \in V_{p \neq q}} \frac{e_{p,q}^\top L_Y^+ e_{p,q}}{e_{p,q}^\top L_X^+ e_{p,q}}$, which is upper bounded by $\lambda_{max}(L_Y^+ L_X)$ (Cheng et al., 2021). With this inequality, the largest generalized eigenvalues and their corresponding eigenvectors can be leveraged as good surrogates for the maximum DMD. Specifically, the stability of an edge $(p, q)$ can

be quantified by measuring the spectral embedding distance between nodes $p$ and $q$. Denote the first $r$ dominant generalized eigenvectors of $L_X L_Y^+$ by $u_1, u_2, ..., u_r$. Because $(u_i^\top e_{p,q})^2 \approx \alpha_i^2 \gg 0$, we will have $\|V_r^\top e_{p,q}\|_2^2 \propto (\delta^F(p,q))^3$ (Cheng et al., 2021).

By computing the average effective-resistance distance between node $p$ and all its adjacent nodes on manifolds, the average DMD for data sample $p$ can be represented as(Cheng et al., 2021):

$$\text{\bf Node DMD score}: \frac{1}{|\mathbb{N}_X(p)|} \sum_{q_i \in \mathbb{N}_X(p)} \left( \|V_N^\top e_{p,q}\|_2^2 \right) \propto \frac{1}{|\mathbb{N}_X(p)|} \sum_{q_i \in \mathbb{N}_X(p)} \left( \delta^M(p,q_i) \right)^3 \quad (4)$$

where $\mathbb{N}_X(p)$ denotes the set of neighbors of node $p$ in the input graph manifold $G_X$. $V_N$ is the generalized eigenvectors of $L_Y^+ L_X (L_X/L_Y$ is the Laplacian matrix of input/output graph). More details are available in Appendix A.8.

## 3 OVERVIEW OF THE PROPOSED SAGMAN FRAMEWORK

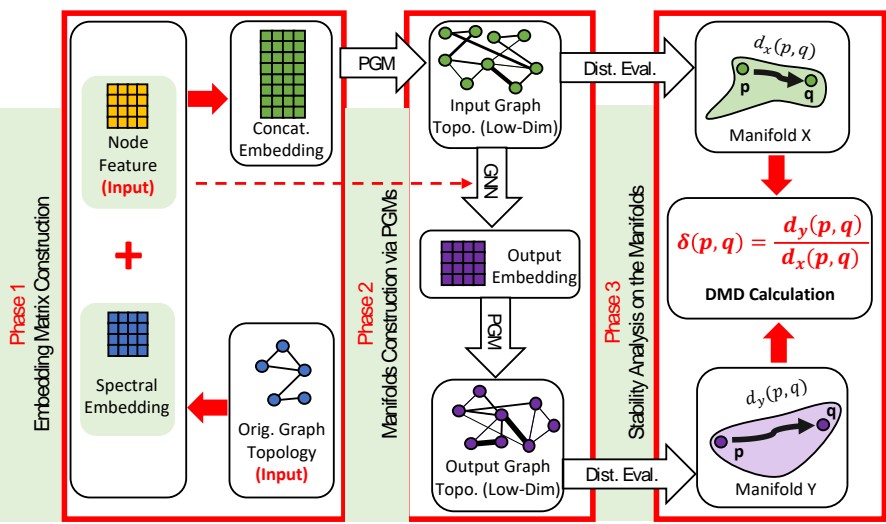

Figure 1: The proposed SAGMAN framework for stability analysis of GNNs on the manifolds.

The rapidly increasing application of GNNs in various domains has underscored the necessity to comprehend and ensure their stability. Stability in the context of GNNs refers to the stability of the network in preserving the structural and topological properties of a graph, even when subject to minor perturbations (Gama et al., 2020). This stability, especially viewed through the lens of individual nodes, serves as a pivotal criterion for the reliable performance of a GNN. Naturally, the stability qualifications employed in the broader realm of neural networks can be repurposed to assess the stability of GNNs.

Nevertheless, applying DMD to GNNs presents several considerable challenges. Firstly, DMD hinges on the assumption that data can be faithfully represented within a low-dimensional manifold. This presumption may not always hold true, particularly in complex data structures handled by GNNs. This distinctive aspect further complicates the direct application of DMD to GNNs. Consequently, these challenges necessitate a more nuanced approach to effectively utilize DMD in the context of GNNs.

To apply the DMD to GNNs, in this work, we leverage PGMs to create low-dimensional manifolds. Figure 1 is the overview of our SAGMAN approach. Our methodology is organized into three distinct phases. In the first phase, we construct an embedding matrix by exploiting node features as well as spectral graph embedding to retain the original graph's key (structural) characteristics. The second phase involves the creation of low-dimensional input and output graph-based manifolds leveraging a highly scalable PGM construction algorithm. The last phase quantifies the distance mapping distortions that encapsulate the mappings (via GNNs) between the input and the output

manifolds. In the following sections, we delve into the specifics of our approach to graph dimension reduction via PGMs.

In Section 4, we highlight the limitations of previous work and underscore the contribution of the proposed approach. Our scalable SAGMAN kernels are detailed in Section 5. Section 5.3 explores the link between the constructed and original graphs. Finally, a thorough analysis of SAGMAN's computational complexity is presented in Section 5.4.

## 4  DMD CALCULATIONS IN SAGMAN

**Challenges in Computing DMDs for GNN Applications.** It is crucial to note that Equation 4 assumes both input and output data lie in low-dimensional manifolds (Fefferman et al., 2016). While the assumption typically holds on GNN outputs (i.e., node embedding vectors), it is often invalid for the input graphs, which reside in a relatively high-dimensional space (Bruna et al., 2013). This poses a key challenge in accurately quantifying sample stability using the DMD metric, as demonstrated by the experiment results in Appendix A.5.

**Graph Dimensionality Reduction via Laplacian Eigenmap.** Graph dimension is defined as the minimum integer that permits a 'classical representation' of the graph within an Euclidean space ($\mathbb{R}^n$) while preserving the unit length of all edges. The dimensionality of a graph is inherently restrained by the Euclidean dimension (Soifer, 2009). However, accurately quantifying either the graph dimension or the Euclidean dimension presents a computational challenge, as it is an NP-hard problem (Schaefer, 2012). Our approach (SAGMAN) for graph dimensionality reduction is based on the well-known nonlinear dimensionality reduction algorithm *Laplacian Eigenmap* (Belkin & Niyogi, 2003). The key idea behind Laplacian Eigenmaps is to first construct a nearest-neighbor (NN) graph of the original data (high-dimensional vectors) and then exploit the first few Laplacian eigenvectors to map each data sample (node) into a low-dimensional vector representation that can be subsequently used in many downstream machine learning tasks, such as clustering, classification, and visualization, etc. To apply Laplacian Eigenmap for dimensionality reduction of a given graph, a naive approach is to first compute an $N$-dimensional vector representation for each node using the complete set of graph Laplacian eigenvectors/eigenvalues by following a spectral graph embedding procedure (Ng et al., 2001), where $N$ denotes the number of nodes; next, the steps in Laplacian Eigenmap can be applied to the embedding matrix obtained in the previous step. To avoid the expensive procedure for computing all Laplacian eigenvectors/eigenvalues, SAGMAN exploits large eigengaps within the first few graph Laplacian eigenvalues which can serve as a valuable metric for encapsulating the global structure of the original graph. The existence of a lower bound for a significant eigengap has been theoretically established as $\Upsilon(k) = \frac{\lambda_{k+1}}{\rho(k)}$ (Peng et al., 2015), where $\rho(k)$ represents the $k$-way expansion constant, and $\lambda_{k+1}$ is the $(k+1)$-th smallest eigenvalue of the normalized Laplacian matrix. Notably, quantifying $\rho(k)$ with precision is challenging, and "large" remains subjective. However, SAGMAN does not necessitate the precise computation of graph dimensions. The identified gap serves as an indicative measure of the approximate graph dimension shifts. This gap is also a way to assess the applicability of SAGMAN to a given graph. Importantly, in spectral graph theory, a large eigengap at $k$ suggests that the embedded distances associated with $k$ smallest eigenpairs and the $k + 1$ smallest eigenpairs are closely comparable (Ng et al., 2001) (Additional discussions are available in Appendix A.2). Empirically, given the dataset class number $c$, $k$ can be approximated using $k \approx 10c$ to encompass the pronounced eigengap (Deng et al., 2022).

## 5  PGM CONSTRUCTION IN SAGMAN

To encode high-dimensional data associated with nodes, edges, or (sub)graphs into low-dimensional representations that well preserve the original graph structural (low-dimensional manifold), we construct PGMs (Feng, 2021). This approach leverages Laplacian Eigenmaps ensuring the retention of the smooth, low-frequency information, and preservation of global structures in the graph (Belkin & Niyogi, 2003). However, the prior state-of-the-art method SGL (Feng, 2021) has its drawbacks. It may require a significant number of iterations to converge, resulting in a worst-case time complexity of $O(N^2 \log^2 N)$. This high computational demand can limit SGL's utility for constructing large graphs, indicating the necessity for more efficient methodologies.

**Gradient-based Scheme for PGM Construction.** In this work, our low-dimensional manifold approach aims to preserve important spectral information through PGMs. Consider a weighted undirected graph $G = (V, E, w)$, with edge weights $w \in \mathbb{R}_{\geq 0}^{V}$ and $|V| = N$. We expand the Laplacian matrix of $G$ as follows:

$$L = \sum_{(p,q) \in E} w_{p,q} e_{p,q} e_{p,q}^{\top} \tag{5}$$

where $e_{p,q} = e_p - e_q$, and $w_{p,q}$ denotes the weight of the edge $(p, q)$. $e_p \in \mathbb{R}^N$ denotes the standard basis vector with all zero entries except for the $p$-th entry being 1. We compute the ascending Laplacian eigenvalues by $\lambda_i$ for $i = 1, ..., N$, and expand the objective function $F = F_1 - \frac{1}{M} F_2$ in Equation 1 as follows:

$$
\begin{aligned}
F_1 &= \log \det(\Theta) = \sum_{i=1}^{N} \log(\lambda_i + \frac{1}{\sigma^2}) \\
F_2 &= Tr(X^{\top} \Theta X) = \frac{Tr(X^{\top} X)}{\sigma^2} + \sum_{(p,q) \in E} w_{p,q} \| X^{\top} e_{p,q} \|_2^2.
\end{aligned}
\tag{6}
$$

Taking partial derivatives with respect to $w_{p,q}$ leads to(Zhang et al., 2022):

$$
\begin{aligned}
\frac{\partial F_1}{\partial w_{p,q}} &= \sum_{i=1}^{N} \frac{1}{\lambda_i + 1/\sigma^2} \frac{\partial \lambda_i}{\partial w_{p,q}} = R_{p,q}^{eff} \\
\frac{\partial F_2}{\partial w_{p,q}} &= \| X^{\top} e_{p,q} \|_2^2 = D_{p,q}^{data},
\end{aligned}
\tag{7}
$$

where $R_{p,q}^{eff}$ and $D_{p,q}^{data}$ denote the effective-resistance distance and $\ell_2$ distance between nodes $p$ and $q$, respectively.

**PGM Refinement via Graph Sparsification.** To maximize the objective function, we prune the dense initial graph by removing the non-critical edges(edges whose addition, removal, or modification does not significantly impact the spectral properties of the graph) with small effective-resistance distance ($R_{p,q}^{eff}$) and large $\ell_2$ distance ($D_{p,q}^{data}$) between nodes p and q. This approach maximizes $F_2$ without significantly decreasing $F_1$. We demonstrate that this pruning strategy is equivalent to pruning edges with small distance ratios, defined as:

$$\rho_{p,q} = \frac{R_{p,q}^{eff}}{D_{p,q}^{data}} = w_{p,q} R_{p,q}^{eff} \tag{8}$$

Since the $w_{p,q} R_{p,q}^{eff}$ is the edge sampling probability for spectral graph sparsification (Spielman & Teng, 2011). Our pruning strategy targets edges with minimal $\rho_{p,q}$. Removing these edges maximally decreases $F_2$ without causing a substantial reduction in $F_1$ in Equation 6. **Consequently, constructing PGMs can indeed be obtained by pruning a dense graph through spectral graph sparsification.**

## 5.1 DENSE GRAPH CONSTRUCTION

To construct a dense graph, it is first necessary to acquire data that predominantly exhibits low-dimensional characteristics. This is attainable through Laplacian Eigenmaps (Belkin & Niyogi, 2003). We define the weighted spectral embedding matrix as Equation 3. In Appendix A.2, we delve deeper into the rationale behind our weighted spectral embedding matrix definition. This approach allows us to maintain the graph's essential topological and spectral properties. Furthermore, node features can be combined with dominant eigenvector components for the construction of the kNN graph (Deng et al., 2022). Subsequently, the k-nearest-neighbor (kNN) algorithm can be employed for the construction of the dense graph. Then, the $O(|V| \log |V|)$ computational complexity kNN algorithm (Malkov & Yashunin, 2018) can be effectively utilized to construct the dense graph.

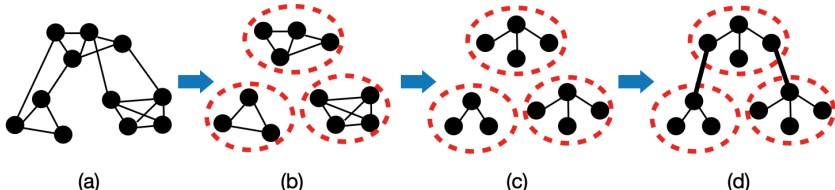

Figure 2: The proposed graph spectral sparsification based on short-cycle decomposition. (a) The original graph with 18 edges. (b) The proposed low-resistance-diameter (LRD) decomposition scheme. (c) The LSSTs within clusters for reducing the number of non-critical edges. (d) The sparsified graph with only 10 edges for well preserving key spectral (structural) properties.

## 5.2 SPECTRAL SPARSIFICATION VIA SHORT-CYCLE DECOMPOSITION

Solving Eigenpairs, a requisite for spectral sparsification (Spielman & Srivastava, 2008), is computationally costly. Yet, employing short-cycle decomposition algorithms for spectral graph sparsification tasks has been shown to maintain near-linear complexity, offering a practical solution(Chu et al., 2020). Short-cycle decomposition partitions an unweighted graph $G$ into multiple disjoint cycles by removing a fixed number of edges, while ensuring a bound on the length of each cycle. Recent graph sparsification algorithm (Lemma 1) combines short-cycle decomposition with low-stretch spanning trees (LSSTs) to construct the sparsified graph for preserving the spectral properties of the original graph (Liu et al., 2019).

**Lemma 1.** *Spectral sparsification of an undirected graph $G$ with its Laplacian denoted by $L_G$ can be obtained by leveraging a short-cycle decomposition algorithm, which returns a sparsified graph $H$ with its Laplacian denoted by $L_H$ such that for all real vectors $x$, $x^\top L_G x \approx x^\top L_H x$ (Chu et al., 2020).*

As a substantial extension of prior short-cycle-based algorithms, we introduce an improved spectral sparsification algorithm. Our method is based on a low-resistance-diameter (LRD) decomposition scheme which aims at restricting the length of each cycle measured with effective-resistance metric. Unlike the prior works, the proposed algorithm is suitable for sparsifying general weighted undirected graphs. The key idea in our method is to efficiently compute the effective-resistance of each edge and employ a multi-level framework to decompose the graph into several disjoint cycles bounded by an effective-resistance threshold.

Figure 2 depicts the application of the short-cycle decomposition method in spectral graph sparsification. The original graph is first decomposed into multiple disjoint node clusters utilizing the proposed LRD decomposition method. Note that the length of each cycle (node cluster) can be adjusted according to a predefined effective-resistance diameter threshold for controlling graph sparsification (spectral approximation) level.

## 5.3 CONSTRUCTED GRAPH AND THE ORIGINAL GRAPH

The DMD establishes a connection between effective-resistance distances and stability analysis (Cheng et al., 2021). For an undirected and connected graph, Feng (2021) demonstrates that the distance between any two nodes $p$ and $q$ can be accurately computed using nontrivial weighted Laplacian eigenvectors (detailed insights are elaborated in Appendix A.2). As discussed in Section 4, employing a suitable number of smallest eigenpairs allows our approach to efficiently approximate the original distances. This is further substantiated in Appendix A.10, where we show that the graph constructed using our method adeptly approximates the effective-resistance distances in the original graph, providing detailed insight and discussion.

## 5.4 COMPLEXITY OF SAGMAN

For spectral graph embedding, we can exploit fast multilevel eigensolvers that allow computing the first $c$ Laplacian eigenvectors in nearly-linear time $O(c|V|)$ without loss of accuracy (Zhao et al., 2021). Then, the $k$-nearest neighbor algorithm (Malkov & Yashunin, 2018) computational complexity is $O(|V|\log|V|)$. Sparsification via short-cycle decomposition has $O(|V|dm)$ time complexity. By leveraging fast Laplacian solvers(Zhao et al., 2021), total nodes' DMD can be computed with $O(|E|)$,

where the $V/E$ denotes the number of nodes/edges, $d$ is the average degree of the matrix, and m is the order of Krylov subspace. SAGMAN's algorithm flow is in Appendix A.4. The SAGMAN was run on the ogbn-arxiv dataset, completing in 174.39 seconds. The dataset comprises 169,343 nodes and 1,166,243 edges. Experiments were conducted on a system with an i9 10900kf processor, 32GB RAM, and an NVIDIA Titan RTX 24GB GPU.

## 6 EXPERIMENTS

We present elementary numerical experiments to illustrate the efficiency of our metric for quantifying the stability of GNNs against node features and edge perturbation conducted on both homophilic and heterophilic datasets. Then, we further show the effectiveness and efficiency of the SAGMAN-guided graph adversarial attack. Additional experiments demonstrating the efficiency of our metric in reducing graph dimensions can be found in Appendix A.2.

**Experimental Setup**. See Appendix A.7 for dataset details. We employ backbone GNN models such as GCN (Kipf & Welling, 2016), GPRGNN (Chien et al., 2020), GAT (Veličković et al., 2017), APPNP (Gasteiger et al., 2018), and ChebNet (Defferrard et al., 2016). Perturbations include Gaussian noise evasion attacks and adversarial attacks (DICE (Waniek et al., 2018), Nettack (Zügner et al., 2018), FGA (Chen et al., 2018)). The input manifold relies on graph adjacency and node features, while the output manifold is constructed from post-*softmax* vectors. To assess GNN local stability, we employ SAGMAN to sample $1\%$ of the entire dataset for stability evaluation. This decision stems from that only a portion of the dataset significantly impacts model stability (Cheng et al., 2021; Hua et al., 2021; Chang et al., 2017). Additional evaluation results on the entire dataset can be found in Appendix A.11. We quantify output perturbations using cosine similarity and Kullback-Leibler divergence (KLD). Additional insights on cosine similarity, KLD, and accuracy can be found in Appendix A.1. For large-scale dataset ogbn-arxiv, due to its higher output dimensionality, we focus exclusively on accuracy comparisons. This decision is informed by the KLD estimator's $n^{-\frac{1}{d}}$ convergence rate (Roldán & Parrondo, 2012), where n is the number of samples and d is the dimension.

### 6.1 STABILITY EVALUATION

Intuitively, when GNNs are locally stable/unstable near certain samples (nodes), these samples' original output and perturbed output should exhibit higher/lower similarities. For example, nodes near locally stable should result in higher cosine similarities. On the other hand, the KLD measures the divergence between two probability distributions. Therefore, nodes near locally stable should yield lower KL divergence under perturbations.

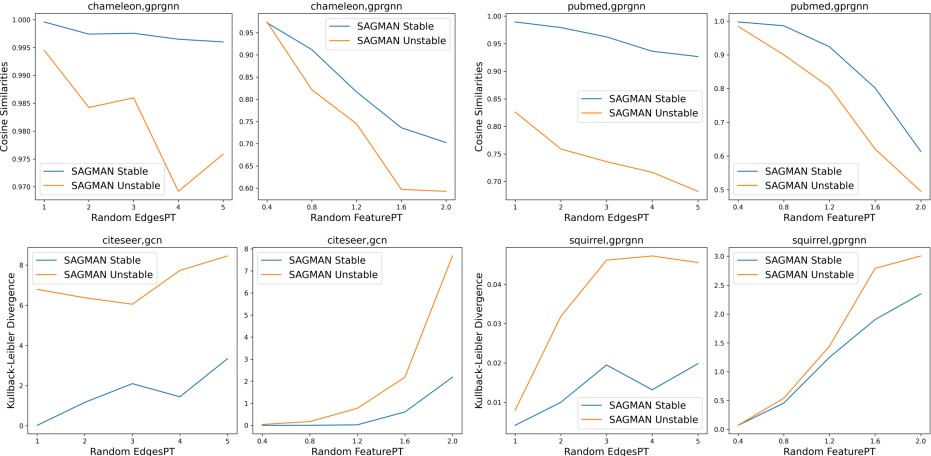

Figure 3: The horizontal axes denote the level of perturbation. 'Random FeaturePT' corresponds to the DICE adversarial attack. 'Random FeaturePT' represents Gaussian noise perturbation $X + \xi\eta$, where $X$ is the feature matrix, $\eta$ is Gaussian noise, and $\xi$ is the noise perturbation level. The upper/lower figures are cosine similarity/KLD.

We demonstrate the capability of SAGMAN for distinguishing between local stable and local unstable in Figure 3. Obviously, SAGMAN is able to select truthfully local stable/unstable that exhibit less/more significant output perturbation. More results regarding different dataset, GNNs, and Nettack attacks are available in Appendix A.7. It illustrates our metric proves its effectiveness. Moreover, We demonstrate stability measurements without SAGMAN in Appendix A.5, which illustrates that DMD calculations can not meaningfully quantify stability when directly using the original input graph structure. Last, we report accuracy differences regarding large-scale datasets in Table 1.

Table 1: SAGMAN selected Ogbn-arxiv samples near GNNs locally stable/unstable. Corresponding accuracies in GCN under different Gaussian Noise Perturbation $X + \xi\eta$, where $X$ is the feature matrix, $\eta$ is Gaussian noise, and $\xi$ is the noise perturbation level.

| Gaussian Noise Perturbation | Accuracy (Stable/Unstable) |
|---|---|
| Clean | 0.91/0.80 |
| Noise Level 1 | 0.91/0.61 |
| Noise Level 2 | 0.89/0.43 |
| Noise Level 3 | 0.89/0.29 |
| Noise Level 4 | 0.87/0.20 |

## 6.2 SAGMAN-GUIDED ADVERSARIAL TARGETED ATTACK

Our GNN training methodology is adapted from the approach delineated by Jin et al. (2021). We adopt GCN as our base model for Citeseer, Cora, Cora-ml, and Pubmed, with Nettack serving as the benchmark attack method. For comparison, target node selection follows Nettack's recommendation (Zügner et al., 2018), SAGMAN-guided strategies, and the heuristic confidence ranking (Chang et al., 2017). Table 2 shows error rates after Nettack attacks and FGA attacks. It shows that the SAGMAN-guided attack outperforms both Nettack node selection and confidence ranking, leading to more effective targeted attacks of both Nettack and FGA. Furthermore, SAGMAN-guided FGA (Chen et al., 2018) also demonstrates superior effectiveness.

Table 2: Error rates comparisons for 40 nodes selected by Nettack's recommendation, heuristic confidence, and SAGMAN-guided strategies. All nodes chosen by SAGMAN are correctly classified prior to perturbation. We bold better error rates.

| Dataset | Nettack Error Rate | Nettack Confidence Ranking Error Rate | SAGMAN-Guided Nettack Error Rate |
|---|---|---|---|
| Cora | 0.725 | 0.925 | **0.975** |
| Cora-ml | 0.750 | 0.800 | **0.950** |
| Citeseer | 0.800 | 0.925 | **1.000** |
| Pubmed | 0.750 | 0.750 | **0.825** |
| Dataset | FGA Error Rate | FGA Confidence Ranking Error Rate | SAGMAN-Guided FGA Error Rate |
| Cora | 0.850 | 0.775 | **0.975** |
| Cora-ml | 0.850 | 0.700 | **0.950** |
| Citeseer | 0.875 | 0.950 | **0.975** |
| Pubmed | 0.875 | 0.925 | **0.950** |

## 7 CONCLUSIONS AND FUTURE WORKS

In this paper, we introduce a novel method for assessing the stability of Graph Neural Networks (GNNs). Our approach devises a metric score reflecting each input data sample's stability level. By evaluating the DMDs on carefully constructed low-dimensional manifolds, we identify graph edges susceptible to perturbations. Aggregating the DMD scores for edges associated with each sample, we present a scalable method for assessing GNN stability in node-level tasks. Our approach provides a comprehensive framework for understanding and quantifying GNN stability, aiding in the enhancement of targeted attacks. Future work will explore improving model stability and reducing robust training costs with SAGMAN.

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
