# OpenReview forum: "SAGMAN: Stability Analysis of Graph Neural Networks (GNNs) on the Manifolds"
_ICLR.cc/2024/Conference — ICLR 2024 Conference Withdrawn Submission_

### Official Review · Reviewer_yWiM · 2023-10-22

**Soundness:** 4 excellent
**Presentation:** 2 fair
**Contribution:** 3 good
**Rating:** 6
**Confidence:** 3

**Summary:**

This paper pioneers a spectral approach to ascertain the stability of GNNs, a critical step towards the dependable application of these networks across diverse domains. The innovative method expands on SPADE's framework by computing node-level stability, leveraging distance mapping distortions on input/output manifolds. The authors' strategy in employing Proximity Graph Manifolds to streamline input dimensions effectively preserves integral structural attributes, marking a significant stride in this field of research.

**Strengths:**

S1: The paper breaks new ground in the realm of GNN application by introducing the first-ever node-level model that quantifies GNN stability. This advancement fills a significant gap in the literature and provides a foundation for further explorations into GNN reliability, an aspect vital for the broader adoption of GNNs in high-stakes scenarios. \
S2: The experimental results are quite promising. \
S3: The near-linear time complexity of the proposed method stands out, particularly in a field where computational overhead can often be prohibitive. Furthermore, the model's agnostic nature concerning specific GNN architectures underscores its versatility and broad applicability, a notable advantage in the rapidly evolving landscape of neural networks. \
S4: The utilization of PGMs for dimension reduction while maintaining key structural characteristics is commendable. It underscores a methodological rigor in ensuring that the stability assessment is grounded in the preservation of graph properties that are most pertinent to GNNs’ functioning.

**Weaknesses:**

W1: The paper would benefit significantly from a more explicit comparative analysis with SPADE. Given that SPADE can be adapted to a range of machine learning models, elucidating the specific enhancements introduced with SAGMAN, particularly those that are GNN-centric, would provide clearer justification for this novel approach. The extension beyond SPADE is one of the paper’s core contributions, and as such, a detailed delineation of the improvements over predecessor models is paramount. \
W2: The narrative in Section 4 could be enhanced for readability and comprehension. The authors are encouraged to consider relocating pertinent content from Appendix A.5 into the main body of the paper to underscore the PGM's critical role. Furthermore, the "Graph Dimensionality Reduction via Laplacian Eigenmap" segment warrants a more lucid exposition. The current presentation may obfuscate the novel methodological insights for readers not deeply versed in this niche area. \
W3: The paper could articulate more clearly the rationale behind key decisions, such as the use of graph dimension reduction. Elaborating on why this step is crucial, specifically in the context of GNNs, would strengthen the readers' appreciation of the method's design. Similarly, a clear, step-by-step walkthrough of the use of Laplacian Eigenmap in this process, supplemented perhaps by schematic diagrams or illustrative examples, would greatly enhance the section's clarity and impact.

**Questions:**

Q1: The text in Figure 1 is garbled. Such elementary mistakes should not occur. \
Q2: Could the authors elaborate on the distinctions between the newly proposed SAGMAN and the existing SPADE framework (Cheng.et.al 2021)? What specific gaps or limitations in SPADE prompted the development of SAGMAN, and how does the latter address these issues more competently, especially in the context of GNNs? \
Q3: The use of graph dimension reduction is a pivotal point in the paper's methodology. Could the authors clarify its necessity and elucidate the specific role and implementation of the Laplacian Eigenmap in this process?

---

### Official Review · Reviewer_KuBc · 2023-11-01

**Soundness:** 3 good
**Presentation:** 3 good
**Contribution:** 3 good
**Rating:** 5
**Confidence:** 3

**Summary:**

This paper provides a framework to evaluate the stability of GNN models. This framework is based on the evaluation of distance mapping distortions between low-dimensional manifolds corresponding to the input and output.

**Strengths:**

This paper addresses a relevant problem of investigating the stability of GNN models at the node-level using SAGMAN framework. In contrast to many existing (theoretical) works in this domain, SAGMAN is data-centric and generic enough to investigate stability of different GNN models in the literature. Furthermore, SAGMAN achieves nearly linear computational complexity and therefore, is scalable to large graphs.

**Weaknesses:**

- It would have been informative to compare the conclusions drawn by SAGMAN regarding the stability of GNNs with the comparable theoretical results (Gama et al., 2020 and Kenlay et al., 2021 cited in the paper, for instance). Although this comparison might be limited to the specific graph constructions and perturbations studied in these theoretical results, it will still add confidence to the soundness of this approach.

- Technical discussions on the limits of PGM construction and graph sparsification in SAGMAN  are missing. Such discussions are necessary to provide the reader more context to gauge the applicability of SAGMAN in practice.

- Experiments: The labels 'SAGMAN stable' and 'SAGMAN unstable' are not explicitly defined. Hence, the interpretations to be made from the experiment results are not clear to me.

**Questions:**

Existing approaches in the domain of explainability of GNNs (see [*] for a review of such methods) rely on the relative variations in the output with respect to the perturbations in the input to determine the importance of input features to the statistical outcome. In this context, how can you distinguish the scenario where the GNN is unstable to specific alterations in the graph structure from the scenario where the specific graph structure may be instrumental to the inference outcome by the GNNs?



[*] Yuan, Hao, et al. "Explainability in graph neural networks: A taxonomic survey." IEEE transactions on pattern analysis and machine intelligence 45.5 (2022): 5782-5799.

---

### Official Review · Reviewer_qUi4 · 2023-11-04

**Soundness:** 2 fair
**Presentation:** 3 good
**Contribution:** 2 fair
**Rating:** 3
**Confidence:** 3

**Summary:**

In this paper, in order to create low-dimensional input/output manifolds for meaningful distance mapping distortions (DMD) estimations while exploiting both the input graph topology and node features, the authors propose a spectral sparsification framework for estimating probabilistic graphical models (PGMs) such that the constructed input/output graph structures can well preserve pair-wise distances on the manifolds. Numerical experiments show that SAGMAN can effectively reveal each node’s stability under various edge/feature perturbations.

**Strengths:**

+ The paper is well-written, clear, and has illustrative figures.
+ This paper is a novel contribution that broadens the understanding of stability of GNNs.
+ The wide applicability could lead to high impact.

**Weaknesses:**

- Missing the standard deviation of accuracy/error rate of the proposed SAGMAN model in Tables 1 and 2.
- Weak empirical evaluation. I appreciate that the authors provide running time of SAGMAN in Section 5.4. Can the authors also provide the running time comparison with other baselines?
- How the authors select the hyperparameters? Sensitivity analysis is missing.
- I wonder how the authors define weights in citation networks?
- Can the authors consider other attack strategies such as Mettack, PGD attack, etc.

**Questions:**

Please see comments in the Weaknesses.

**Details Of Ethics Concerns:**

Not applicable.

---

### Official Review · Reviewer_9cB9 · 2023-11-06

**Soundness:** 3 good
**Presentation:** 2 fair
**Contribution:** 2 fair
**Rating:** 3
**Confidence:** 4

**Summary:**

The paper uses DMD as a notion of the stability of a GNN, which is a mapping of a graph to another graph. The node feature of the input graph consists of the original node features and spectral embedding of the nodes. The output graph's node features are the embedding via the GNN. Sparsification is used to speed up the computation of the spectral embeddings.

**Strengths:**

1) the use of DMD notion and the related computational technique is interesting;
2) positive experimental results under regualr and attacking situations are promising;
3) the challenge of graph neural network stability is meaningful to address.

**Weaknesses:**

1) the definition and the construction of PGM is not formally and clearly stated
2) how the improved stability is obtained is unclear.
3) the work largely use existing techniques, while how the techniques are combined to address the challenges is unclear.

**Questions:**

1) the M in the definition of X on page is not defined.
2) what's the relationship between equations (2) and (3)? There are cited references but the authors shall make the paper self-contained.
3) What's the difference between \delta^F and \delta^M?
4) Why the defined PGM can create a low-dimensional manifolds? I suggest the authors refer to the book

"Probabilistic Graphical Models: Principles and Techniques" by Koller

for a more commonly used definition of PGM. I am not sure if "PGM" is the term what the authors intended to use here.
5) How to evaluate the derivatives of the eigenvalues with respect to w?
6) the proposed method below Lemma 1 lacks of explanations and justification.
7) In the experiment, the authors employ existing GNN models, then where does the improved stability come from? Is it due to the pruning and sparsification? If so, pruning and sparsification come from existing work and what's authors' novel contributions?

---

### Official Review · Reviewer_eSPk · 2023-11-09

**Soundness:** 2 fair
**Presentation:** 2 fair
**Contribution:** 2 fair
**Rating:** 5
**Confidence:** 3

**Summary:**

This paper proposed novel methods for evaluating the stability of GNNs. To measure how much the outputs of GNNs vary when varying the input features, the proposed methods construct graphs by utilizing DMD and PGM. Then, this paper demonstrated that the proposed methods can properly capture the stability of GNNs by evaluating how much the outputs of GNNs vary when adding noise to the input features.

**Strengths:**

This paper proposed the methods to measure the stability of GNNs. The experiments demonstrated that the proposed methods can measure the robustness of noise added to the input features. They also showed that the proposed methods can be applicable to select weak nodes for adversarial attacks.

**Weaknesses:**

* I felt it was difficult to understand the proposed methods due to the presentation issues. According to Alg. 1, the proposed methods seem to consist of four steps: compute_weighted_spectral_embedding, construct_kNN_graph, sparse_graph, and get_low_dimensional_graph, and Sec. 5 tries to explain these steps. However, without looking at Alg. 1 shown in Appendix, it would be difficult to get an overview of the proposed methods from the main paper and Fig. 1 only.
* The detailed experimental settings are lacking.
    * It is not explained how the authors tuned hyperparameters for proposed methods and GNNs.
    * It is unclear what "SAGMAN stable" and "SAGMAN unstable" indicate in Fig. 3 and Table 1. Did the authors split nodes into stable and unstable nodes by the proposed methods?
    * It is unclear how the authors measure cosine similarity and Kullback-Leibler divergence in Fig. 3. Did the authors measure the distance between outputs of the last layer of GNNs in the case with and without noise?

Minor Comments:
* In my environment, the texts in Fig. 1 (e.g., Phase 1 Embedding Matrix Construction) are garbled, and I could not read them. Finally, I could read it by changing my browser from Firefox to Chrome, but it would be helpful if the authors could check the character code, etc.
* There are several typos. There must be a space before the quotation or parentheses, e.g., "leads to(Zhang et al., 2022)," "non-critical edges(edges whose addition," "a practical solution(Chu et al., 2020)," "fast Laplacian solvers(Zhao et al., 2021)," and "represented as(Cheng et al., 2021)."

**Questions:**

Please see the weakness section.